# Preserved intention understanding during moral judgments in schizophrenia

Lisa Kronbichler[1,2,3]*, Renate Stelzig-Schöler[3], Melanie Lenger[4], Stefanie Weber[3], Brandy-Gale Pearce[3], Luise-Antonia Reich[5], Wolfgang Aichhorn[3], Martin Kronbichler[1,2]

1 Neuroscience Institute, Christian-Doppler University Hospital, Paracelsus Medical University, Salzburg, Austria, 2 Centre for Cognitive Neuroscience and Department of Psychology, University of Salzburg, Salzburg, Austria, 3 Department of Psychiatry, Psychotherapy & Psychosomatics, Christian-Doppler University Hospital, Paracelsus Medical University, Salzburg, Austria, 4 Department of Psychiatry and Psychotherapy, Medical University of Graz, Graz, Austria, 5 Department of Child and Adolescent Psychiatry, Psychotherapy, and Psychosomatics, University Medical Center Hamburg-Eppendorf, Hamburg, Germany

* li.kronbichler@gmail.com

## Abstract

### Introduction

Although there is convincing evidence for socio-cognitive impairments in schizophrenia spectrum disorder (SSD), little evidence is found for deficient moral cognition. We investigated whether patients with SSD showed altered moral judgments in a story task where the protagonist either had a neutral or malicious intention towards another person. This paradigm examined whether SSD relates to altered moral cognition in general or specifically to impaired integration of prior information (such as beliefs) in moral judgments.

### Methods

23 patients and 32 healthy controls read vignettes created in a 2 x 2 design. The protagonist in each story either had a neutral or negative intention towards another person which, as a result, either died (negative outcome) or did not die (neutral outcome). Participants rated the moral permissibility of the protagonist's action. Standard null hypothesis significance testing and equivalent Bayes analyses are reported.

### Results

Schizophrenia patients did not differ significantly in permissibility ratings from healthy controls. This finding was supported by the Bayes analyses which favoured the null hypothesis. Task performance was not related to symptom severity or medication.

### Conclusions

The current findings do not support the notion that moral judgments are deficient in schizophrenia. Furthermore, the current study shows that patients do not have observable difficulties in integrating the protagonist's belief in the rating of the moral permissibility of the action-outcome.

**Data Availability Statement:** Currently, we do not have approval from the local ethical committee responsible for this study (Ethikkommission des Landes Salzburg - Ethics committee of the regional

state of Salzburg) to publicly share single patients data of any sort, without the explicit consent of patients to share their individual data publicly. Since our study consent form did not include such a clause, we are not allowed to make single-subject data publicly available. Any request for should be addressed to the main assistant of the Head of the Department for Psychiatry, Psychotherapy and Psychsosomatics. She will coordinate the data access after consultation with the ethics commitee Salzburg: Bettina Böhaker Assistant of the Head of Department of Psychiatry, Psychotherapy and Psychosomatics Christian-Doppler Medical Centre Ignaz-Harrer Straße 79 A- 5020 Salzburg Phone: +43 (0)5 7255-35001 E-Mail: b.boehaker@salk.at Ethikkommission für das Bundesland Salzburg Sebastian-Stief-Gasse 2 5020 Salzburg Contact: Mag. Ulrike Wendl-Toiflhart Tel.: +43-(0)662-8042-2375 E-Mail: ethikkommission@salzburg.gv.at.

**Funding:** This work was supported by grants provided to Martin Kronbichler by the Austrian Science Fund (FWF grant number: P 30390-B27) and the Scientific Funds of the Paracelsus Medical University (grant number: E-13/18/097-KRO).

**Competing interests:** The authors have declared that no competing interests exist.

# 1.Introduction

Schizophrenia-spectrum disorders (SSD) are characterized by altered social cognition [1,2], which plays a fundamental role in everyday functional outcome [3,4]. A prominent factor of such socio-cognitive functions is Theory of Mind (ToM), which is the ability to recognize the intentions and beliefs of others and to predict their behavior based on an understanding of their minds [5]. While there is broad agreement in behavioral and neuroimaging research that ToM processes are impaired in schizophrenia [6,7], there is one related aspect that has received relatively little attention and which yields mixed findings, namely moral cognition. Moral cognition describes the thinking about (culture-specific) sets of values and behavior that guide socially accepted manners. It is reflected in research on for example harm and pro- or anti-social behavior [8,9]. On the neuronal level, moral cognition is related to activation in regions that are involved in ToM processes such as TPJ, PFC and precuneus, but also the orbitofrontal cortex and the amygdala [10,11].

Due to their impairments in ToM and empathy, a common assumption is that patients with SSD make more utilitarian choices [12] in moral dilemma tasks. Making utilitarian choices means that one chooses the action that will result in the greatest good for the most, thereby justifying the means (e.g., killing one person to save a group; [13,14]). This idea is supported by the finding that SSD patients show alterations in the prefrontal cortex [7], which is related to moral goal pursuit [15] and that damage in this area is linked to stronger utilitarian thinking [16]. Behavioral studies reveal mixed results, showing fragile effects that often depend on question probe and perspective [14,17–19]. Furthermore, common tasks such as Kohlberg's Moral Judgment Interview require verbal abilities ([20–22]) and abstract reasoning, both of which are impaired in SSD [20–22]. In general, psychiatric research on moral cognition is not yet conclusive. First, the number of available studies is small, thus more research is needed to allow for conclusive meta-analyses. Second, the comparability of published studies is limited since not all experiments examine the same dimension of morality, which is assumed to consist of a) the moral decision itself, b) the moral judgment about how appropriate the action is and c) the moral inference describing how a person is perceived based on his/her action and additional information about the person [23]. Studies on SSD revealed evidence for impaired [13,14,17,22] as well as intact moral decision-making [13,17–19]). There is some evidence for impaired moral judgment in SSD [14,24,25], however, a more recent examination merely found prolonged but otherwise intact moral judgments in patients [14]. Concerning moral inference, there are several studies on healthy controls showing that information about the acting person alters moral permissibility ratings [26–28] and that this process is related to activation in ToM areas [28]. To our knowledge, there is no study on moral inference in SSD available yet. However, prior information about the other person's intention or reliability [29,30] plays a critical role in real-life moral judgment. To illustrate, intentionally killing someone by knowingly putting poisonous granules in the other person's coffee versus unintentionally killing someone by putting granules in their coffee (believing it is *sugar)* are two morally distinct actions, although the physical action and the outcome remain the same in both scenarios (example taken from [29]).

In the current study, we examine how information about the intention of the actor influences how permissibly the action is perceived by healthy participants and patients with SSD. Although we do not directly assess how the actor itself is perceived, an examination of the relationship between prior information about the acting person and moral judgment will provide first insights into the interplay of these two factors.

Since patients with SSD often show deficient mentalizing abilities, we assume that they should reveal problems with integrating the protagonist's belief into their moral judgments

(details below). Evidence for this assumption comes from patients with autism as well as patients with frontotemporal dementia and frontal stroke, who overly rely on action-outcome (utilitarian approach), thus judging accidental harms as less permissible and attempted but failed harms as more permissible [29,31].

Healthy controls and patients with SSD rate the moral permissibility of scenarios that are designed in a 2x2 way: The protagonist in each story can either hold a negative (intention to kill someone) or a neutral belief (no murderous intentions) about his action which then leads to a negative (the other person dies) or neutral outcome (the other person does not die).

If our SSD sample has difficulties understanding the intention and belief of the protagonist, we should find that patients mostly rely their ratings on the action outcome since the intention of the protagonist (to harm or not to harm the other person) is not informative to them.

This should lead to pronounced group differences especially in conditions where the protagonist's intention is in conflict with the situation outcome (e.g. *Accidental Harm Condition*: The protagonist unintentionally kills the other person. *Attempted Harm Condition*: The protagonist aims to kill the other person but does not succeed).

Given that the utilitarian approach holds true for patients with SSD, we expect them to show stricter ratings in the *Accidental Harm* condition and more tolerant ratings in the *Attempted Harm* condition.

However, as mentioned above, moral judgment tasks show mixed results concerning moral impairments in schizophrenia, therefore raising the possibility that patients with SSD might not reveal alterations at all. If this is the case, we will nonetheless provide post-hoc t-tests for each condition, since the critical alterations might be reflected in group differences in single conditions only [29]. Furthermore, we will also include Bayes analyses to tackle the actual support for the null over the alternative hypothesis.

## 2. Methods

Written informed consent was provided by each participant (see below). The study was approved by the local ethics committee *Ethikkommission Salzburg*, Austria (415-E/981/2-2008).

### 2.1 Participants

Participants in the patient group were 23 male adults, who had received a formal ICD-10 diagnosis (which was verified before study participation by certified psychiatrists using the Mini-International Neuropsychiatric Interview [32] in combination with the ICD-10 Diagnostic Checklist for Schizophrenia [33]) in the schizophrenia spectrum group (F20) or the schizoaffective disorders spectrum group (F25). Exclusion criteria for both patients and healthy controls were psychiatric disorders other than schizophrenia or schizoaffective disorders, current or past neurological insults like head trauma, and current substance abuse. Controls were screened for mental and physical health and were excluded if met the criteria for psychiatric disorders or exhibited a family history of psychiatric illness. All patients were recruited from the outpatient and inpatient units of the Department of Psychiatry, Psychotherapy and Psychosomatics, Christian-Doppler Medical Centre, PMU, Salzburg, Austria. All patients received antipsychotic medication. Patients were clinically stable with relatively mild symptoms at the time of assessment (PANSS [34]). Healthy control participants were 32 male adults. Efforts were made to recruit a healthy male control group that matched the SSD group in demographics and education. Thus, advertisements for HCs specified that we were particularly interested in participants who finished high school, but that did not necessarily attend or complete college. Subjects were remunerated for participation and all participants provided written informed consent. Only participants who were authorized to give informed consent

**Table 1. Demographic data and clinical ratings of patients with SSD and controls.**

| | Schizophrenia Patients (n = 23) | Healthy Controls (n = 32) | *p* Value |
|---|---|---|---|
| Age (y) | 25.91 (5.2) | 25.00 (4.3) | .508 |
| EQ | 34.27 (9.9) | 41.8 (10.6) | .012 |
| SCIP | 73.10 (9.9) | 83.31 (7.4) | < .001 |
| MWT | 27.83 (3.2) | 29.13 (3.7) | .185 |
| CPZ (mg) | 409.26 (259.43) | - | - |
| Duration of Illness (y) | 3.75 (4.9) | - | - |
| PANSS+ | 14.37 (6.2) | - | - |
| PANSS- | 15.50 (6.7) | - | - |
| *Education Level** | | | |
| Compulsory | 8% | 22% | - |
| Apprenticeship | 8% | 39% | - |
| A Levels | 64% | 30% | - |
| University degree | 20% | 9% | - |

Note: All variables were examined by independent sample t-tests. Values for Age—PANSS indicate the group mean. Standard deviation of each variable is reported in brackets. EQ = Empathy Quotient; SCIP = Screen for Cognitive Impairment in Psychiatry; MWT = Mehrfach Wortschatz Test (vocabulary test); CPZ = Chlorpromazine equivalent; PANSS = Positive and Negative Syndrome Scale, split in positive and negative symptom subscale

* highest education level achieved. % indicates percentage within each group that reached a certain education level.

themselves (>18 years, not under external custody) were included in the study. Furthermore, patient recruitment was exclusively done by an experienced team of psychiatrists. Demographic data and clinical rating are listed in Table 1.

## 2.2 Socio-cognitive tasks

Moral judgment was examined using a translated (German) paper-pencil version of the moral judgment task [29]. Participants read 24 short vignettes in a 2 x 2 design (intention, outcome). To give an example: Grace and her friend are visiting a chemical plant. Grace gets them coffee and puts a white substance (presumably sugar) into her friend's coffee. The storyline is varied in terms of Grace's intention and the outcome of her action: She can either hold a neutral intention (putting the white substance into her friend's coffee assuming it is sugar) or a negative intention (putting the white substance into the coffee knowing it is poison). Subsequently, her friend either dies (negative outcome) or does not die (neutral outcome). This results in four conditions:

*Accidental Harm* (neutral intention, negative outcome): Grace accidentally puts poison into the coffee believing it is sugar. Her friend dies.

*Neutral Acts* (neutral intention, neutral outcome): Grace puts sugar into the coffee, believing it is sugar. Her friend does not die.

*Attempted Harm* (negative intention, neutral outcome): Grace puts sugar into the coffee, but she believes it is poison. Her friend does not die.

*Intended Harm* (negative intention, negative outcome): Grace knowingly puts poison into the coffee. Her friend dies.

Two full example stories are provided in the supplementary material. Participants rate the moral permissibility (seven-point Likert scale; 1 = completely morally forbidden, 7 = completely morally permissible) of the action. Note that each participant only receives *one* version of each

of the 24 stories. This test exists in four parallel variants which were assigned randomly to our participants and which are described and evaluated in Moran et al (2011).

Since levels of empathy are supposed to relate to utilitarian moral decision-making [35], we examined overall empathy using the Empathy Quotient (EQ) [36] in order to examine the role of empathy during moral judgments in SSD.

### 2.3 Basic cognition

Basal cognitive abilities were estimated using the Screen for Cognitive Impairment in Psychiatry (SCIP) and the multiple-choice vocabulary test (MWT = Mehrfachwahl-Wortschatz-Test).

### 2.4 NHST analyses

A repeated measures ANOVA including the factors *Intention* (neutral x negative), *Outcome* (neutral x negative) and *Group* (SSD x healthy controls) was conducted. Post-hoc t-tests were performed examining group differences for each condition separately. Additionally, Monte-Carlo-Simulation tests were done to examine the replicability of the t-test results. All (but the permutation test) analyses were implemented in JASP (JASP Team, 2018; [37,38]). Monte-Carlo Simulations were conducted using SPSS's exact package. Details are described in the Supplementary Material.

### 2.5 Bayesian analyses

For Bayesian hypothesis testing [39,40] we relied on Bayes Factors (BF). BF reflects the change from prior to posterior odds for specific hypotheses and models given the data. In other words, the BF is the probability of obtaining the data under the alternative hypothesis relative to the probability of obtaining the data under the null hypothesis. In this way, the BF reflects the strength of the evidence for one hypothesis/model compared to another and can also be used to quantify the evidence for the null hypothesis compared to the alternative in light of the data. For example, in Bayesian ANOVAs BFs can reflect the strength of evidence for inclusion compared to the exclusion of different factors and effects from the model [41].

## 3. Results

Mean permissibility ratings are illustrated in Fig 1. Demographic data, medication and cognitive measures are listed in Table 1. Descriptives and results of the ANOVA and t-tests are listed in Table 2.

Pearson Correlations indicate that symptom severity (PANSS) and medication (Chlorpromazine equivalent) were not related to task performance ($rs < .38$, $ps > .09$). Illness duration was moderately related to permissibility ratings in *Intended Harm* scenarios ($r = .41$, $p = .052$).

### 3.1 NHST analyses

**3.1.1 Moral judgment task.** Negative *Intentions* and negative *Outcomes* were rated as significantly less permissible compared to neutral intentions and neutral outcomes, respectively ($Fs(1,53) > 94$, $ps < .001$). These main effects for *Intention* and *Outcome* were qualified by an *Outcome-by-Intention* interaction ($F(1,53) = 24.14$, $p < .001$) since accidental harms were rated as more permissible than intended harms. No main effect of *Group* ($F(1,53) = .03$, $p = .855$) and no interaction with the factor *Group* were observed ($Fs(1,53) < .08$, $ps > .37$). Post-hoc t-tests reveal no significant differences between patients and controls in neither condition (Table 1). The findings of the post hoc t-test are supported by our permutation tests, which do

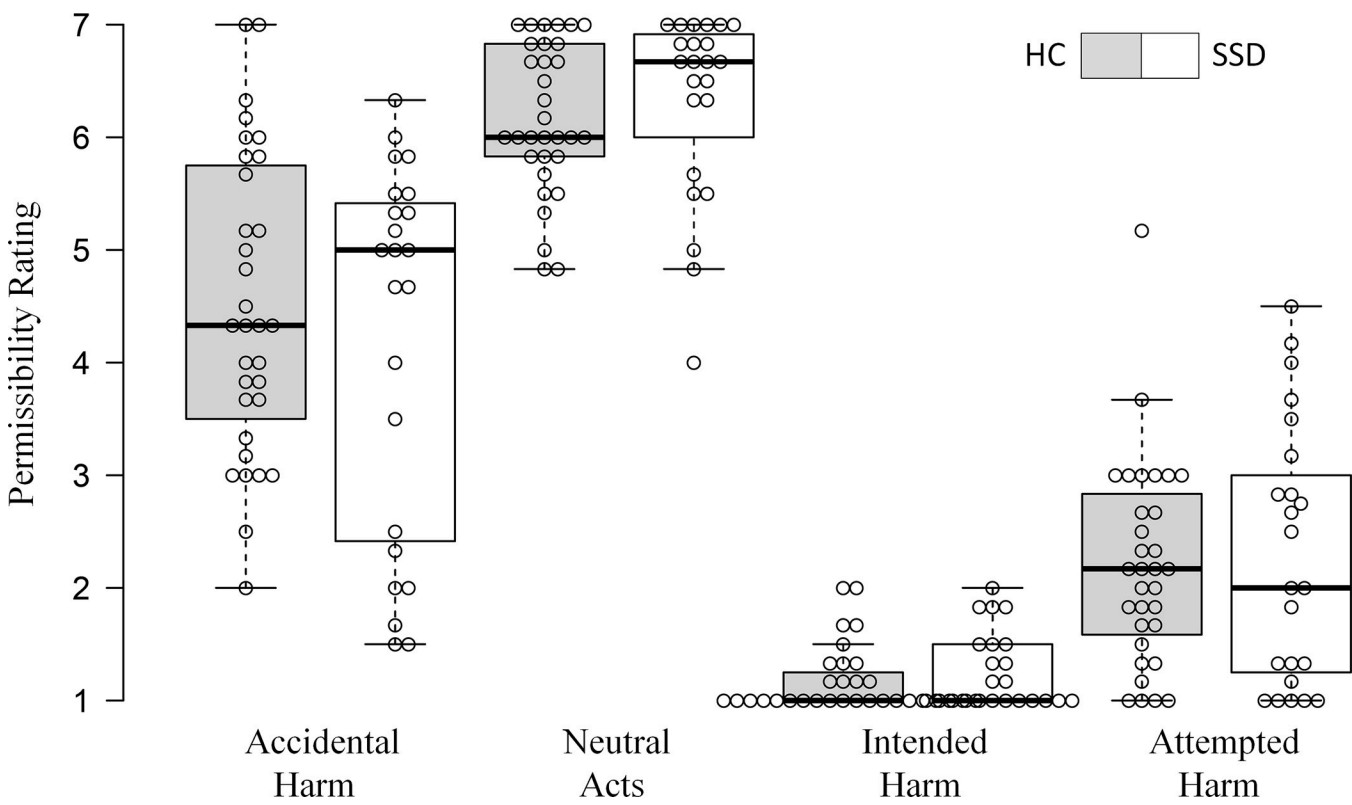

**Fig 1. Tukey Box-plots depict permissibility ratings on a 7-point Likert scale.** Controls (HC) are depicted in grey, patients with SSD in white. Bold horizontal lines indicate the group median, bold crosses show the group mean. End of whiskers indicate the first and third quartile.

not reveal significant group differences in any condition, and which are listed in detail in S1 File. Accordingly, the data do not show evidence for group differences in moral permissibility ratings between healthy controls and SSD patients.

**3.1.2 Additional questionnaires.** Patients with SSD showed significantly decreased performance on the EQ compared to healthy controls (($t$ = 2.6, $p$ = .012). Furthermore, patients scored significantly lower on the SCIP ($t$ = 4.17, $p$ < .001) but showed normal performance on the MWT ($t$ = 1.34, $p$ = .185).

## 3.2 Bayesian analyses

Although our analyses demonstrate a lack of group differences between healthy and SSD participants, this type of analysis does not allow us to examine the extent to which the null hypothesis outperforms the alternative hypothesis (as would be suggested by non-significant group effects). The Bayesian analyses provided in this section offer a promising approach to quantify the support for the null hypothesis [40,42–45].

**3.2.1 Moral judgment task.** A Bayesian Repeated Measures ANOVA with default prior scales including the factors Intention (neutral x negative), *Outcome* (neutral x negative) and *Group* (SSD x healthy controls) was conducted (Table 3). The model that received the most support compared to the null model is the two main effects and interaction model *Intention + Outcome + Intention*Outcome* (BF$_{10}$ = 2858e+74). Adding the factor *Group* would decrease the model support by a factor of 4.5 (2858e74/6327e73), which therefore speaks in favour of the two main effects and interaction model. Overall, there is merely anecdotal support for any model including the factor *Group* or any interaction with *Group* (BFs$_{10}$ < .21). Since our

**Table 2. Descriptives, ANOVA and T-test output values.**

| Descriptives | | | |
|---|---|---|---|
| | **Mean healthy controls** | | **Mean SSD** |
| Accidental Harm | 4.434 (1.3) | | 4.257 (1.6) |
| Neutral Acts | 6.196 (0.9) | | 6.296 (1.2) |
| Intended Harm | 1.167 (0.3) | | 1.272 (0.3) |
| Attempted Harm | 2.208 (0.7) | | 2.269 (0.8) |
| ANOVA | | | |
| | df | F | p |
| Intention | 1, 53 | 94.32 | 0.000 |
| Intention*Group | 1, 53 | 0.148 | 0.702 |
| Outcome | 1, 53 | 458.6 | 0.000 |
| Outcome*Group | 1, 53 | 0.134 | 0.717 |
| Outcome*Intention | 1, 53 | 24.14 | 0.000 |
| Outcome*Intention*Group | 1, 53 | 0.795 | 0.377 |
| Group | 1, 53 | 0.034 | 0.855 |
| Independent Samples (SSD vs. Healthy Controls) T-tests | | | |
| | df | t | p |
| Accidental Harm | 53 | 0.435 | 0.439 |
| Neutral Acts | 53 | 0.486 | 0.629 |
| Intended Harm | 53 | 1.230 | 0.306 |
| Attempted Harm | 53 | 0.219 | 0.485 |

Note: Descriptives: Mean values of 32 healthy controls and 23 patients are listed.

Standard deviation of each variable is reported in brackets. ANOVA: computed using the factors intention (neutral, negative), outcome and group (SSD, controls).

Asterisks indicate interactions.

primary analysis features several candidate models, we examined the change from prior to posterior model plausibility. As evident from Table 4, averaged across all models, our data strongly support the inclusion of the factors *Intention* ($BF_{incl}$ = 8465e13), *Outcome* ($BF_{incl}$ = 8465e13) and *Intention*Outcome* ($BF_{incl}$ = 59.44). Again, weak support is found for models including the factor *Group* ($BF_{incl}$ < .11).

**3.2.2 Additional questionnaires.** Bayesian Independent Samples t-tests reveal very strong evidence for the assumption that there is a difference in performance on the SCIP between controls and patients (BF10 = 137.03). Moderate evidence for a group difference is found for the EQ (BF10 = 6.54), and no (or anecdotal at best) support is evident for differences in the MWT (BF10 = 0.61). Order-restricted (controls > patients; comparable to one-sided t-testing in NHST) t-tests reveal similar findings, elevating the evidence for a group difference in the EQ to moderate-strong (BFs10: SCIP = 273.99; EQ = 13; MWT = 1.09). In sum, Bayes analyses suggest a decreased performance of patients on the SCIP and, slightly less pronounced, on the EQ. No substantial group differences are evident for the MWT.

## 4. Discussion

We examined the performance of patients with Schizophrenia Spectrum Disorder (SSD) on a recent moral judgment task [29] which requires the understanding of others' intentions in order to adequately rate the moral permissibility of actions. We argued in the Introduction that sophisticated moral judgments about other people's behavior requires the integration of their beliefs and intentions. Patients with SSD often show impairments in belief

**Table 3. Bayesian repeated measures ANOVA: Model comparison.**

| Models | P(M\|Data) | BF$_M$ | BF$_{10}$ | error % |
|---|---|---|---|---|
| Null model (incl. subject) | 2.511e -80 | 4.519e -79 | 1.000 | |
| Intention + Outcome + Intention ✻ Outcome | 0.543 | 21.346 | 2.161e +79 | 2.482 |
| Intention + Outcome + group + Intention ✻ Outcome + Intention ✻ group | 0.112 | 2.259 | 4.441e +78 | 2.583 |
| Intention + Outcome + group + Intention ✻ Outcome | 0.084 | 1.660 | 3.363e +78 | 2.894 |
| Intention + Outcome | 0.083 | 1.635 | 3.317e +78 | 1.727 |
| Intention + Outcome + group + Intention ✻ Outcome + Intention ✻ group + Outcome ✻ group | 0.067 | 1.293 | 2.669e +78 | 12.989 |
| Intention + Outcome + group + Intention ✻ Outcome + Outcome ✻ group | 0.043 | 0.815 | 1.724e +78 | 4.108 |
| Intention + Outcome + group + Intention ✻ Outcome + Intention ✻ group + Outcome ✻ group + Intention ✻ Outcome ✻ group | 0.021 | 0.381 | 8.248e +77 | 7.817 |
| Intention + Outcome + group + Intention ✻ group | 0.018 | 0.338 | 7.331e +77 | 4.355 |
| Intention + Outcome + group | 0.013 | 0.240 | 5.246e +77 | 1.641 |
| Intention + Outcome + group + Intention ✻ group + Outcome ✻ group | 0.009 | 0.162 | 3.545e +77 | 4.654 |
| Intention + Outcome + group + Outcome ✻ group | 0.007 | 0.123 | 2.692e +77 | 4.792 |
| Intention | 2.677e -24 | 4.819e -23 | 1.066e +56 | 3.400 |
| Intention + group | 4.101e -25 | 7.381e -24 | 1.633e +55 | 2.232 |
| Intention + group + Intention ✻ group | 2.823e -25 | 5.081e -24 | 1.124e +55 | 5.117 |
| Outcome | 6.708e -74 | 1.207e -72 | 2.672e +6 | 2.813 |
| Outcome + group | 1.187e -74 | 2.137e -73 | 472900.981 | 9.798 |
| Outcome + group + Outcome ✻ group | 2.396e -75 | 4.312e -74 | 95414.483 | 3.270 |
| Group | 3.812e -81 | 6.862e -80 | 0.152 | 0.987 |

Notes: All models include subject. Prior model assignment probability (P(M)) for each model is 0.053. P(M\|data) values indicate the posterior model probability. BF$_M$ describes the change in odds from prior to posterior model. BF$_{10}$ indicates the evidence the data provide for H1 versus H0.

understanding, why we assumed that this task (compared to previous moral judgment tasks which did not require belief integration) might indicate moral judgment abnormalities more robustly. Response patterns of the healthy control participants replicate previous findings

**Table 4. Bayesian repeated measures ANOVA: Analysis of effects.**

| Effects | P(incl) | P(incl\|data) | BF$_{incl}$ |
|---|---|---|---|
| Intention | 0.737 | 1.000 | 5.106e +13 |
| Outcome | 0.737 | 1.000 | 5.106e +13 |
| Group | 0.737 | 0.374 | 0.214 |
| Intention ✻ Outcome | 0.316 | 0.869 | 14.434 |
| Intention ✻ group | 0.316 | 0.227 | 0.635 |
| Outcome ✻ group | 0.316 | 0.147 | 0.372 |
| Intention ✻ Outcome ✻ group | 0.053 | 0.021 | 0.381 |

Notes: P(incl) is the prior factor inclusion probability (The prior probability with which a factor is included in a model). P(incl\|data) is the posterior factor inclusion probability after the data have been observed. BF$_{incl}$ describes the change from prior to posterior inclusion odds.

showing that scenarios with neutral intentions and neutral outcomes are judged as more mor-
ally permissible than negative outcomes and intentions. Accidental harms were rated more
permissible than intended but failed harms, which is again in line with previous findings on
that task [29].

Interestingly, neither the standard ANOVA nor the Bayesian approach show support for
altered moral judgments in patients with SSD but reveal evidence for healthy-like consider-
ations of intentions and outcome despite observable impairments in general cognition and
empathy. In the following, we discuss several reasons which might account for our findings
and discuss their explanatory potential given the data and the current literature.

Probably the most important consideration for a null effect is the statistical power of the
study. Whereas posterior power calculations are under heavy critique [46–48], an a priori
power calculation might have been useful to estimate the necessary number of participants to
detect a true effect. However, a priori power calculations require an estimation of the popula-
tion effect size. Critically, this specific task is not implemented in SSD yet (and the required
Cohens' $f$ does not generalize across task designs) and estimating the effect size in SSD based
on the effect size for another psychiatric population (the original study is based on autistic
patients) appears arbitrary and is beside the point. Furthermore, the effects of interest would
have been reflected in interactions and estimating an ANOVA design in standard power tools
can be tricky (but see [49]): Assuming that SSD patients have difficulties merging the actor's
intention with the situation outcome, they should have rated accidental harms as less permissi-
ble (or attempted harms as more permissible) compared to healthy controls, which both
would have been reflected in a group-by-intention interaction. To nonetheless map the evi-
dential strength of our results given the number of participants, we ran Bayes sequential
robustness analyses which estimate the number of participants after which the Bayes factors
become constant due to a convergence of updated prior distributions to posterior distributions
(for details see [41]. In other words, this analysis shows upon which number of participants
the group differences most likely will remain constant. As evident from S1–S4 Figs in the sup-
plementary material, we reach a constant level of anecdotal (Intentional Harm) and moderate
(Attempted Harm, Neutral Acts, Accidental Harm) support for the null hypothesis in all four
conditions after 40–50 participants in total. Since the overall number of datasets is 55, we do
not assume that there would have been relevant shifts in the overall pattern of results if more
datasets were acquired.

Another reason for the absence of group differences might be that all patients have a com-
parably mild form of SSD or that our sample is biased towards patients with a relatively short
illness duration. A quick descriptive approach of the medical history of our patients shows that
about 35% of our patients have an illness duration of more than 4 years, 17% between two and
three years and 46% are diagnosed for one year. Although most of our patients report personal
changes ~2 years before their first hospitalization, this distribution suggests a high number of
first-episode patients in our sample. Illness severity (as measured with the PANNS) shows a
mean percentile rank of 29 for positive symptoms and a mean rank of 25 for negative symp-
toms. Although our sample scores at the putatively lower range of psychiatric symptoms, they
are in the average range of the norm (schizophrenic) sample. A similar picture emerges for
SPQ and PNS-Q values, where our sample scores at the lower range of comparable psychiatric
samples ([50,51]. Assuming that there exists a true alteration in moral judgments in SSD, the
current data might be explained by either a task insensitivity to detect subtle changes or that
alterations in moral judgment become evident only later in the course of illness or merely
occur in more severe forms of SSD. However, neither symptom severity ($rs(26) < .127$, $ps >
.521$) nor illness duration ($rs(26) < .336$, $ps > .100$) were related to task performance in our

sample and currently available studies on moral judgment in SSD suggest a rather subtle dysfunction even in more severe forms of SSD [21,52].

As shortly mentioned above, an alternative explanation might be that the task is not suitable to detect small alterations in moral judgments in SSD as compared to other clinical samples. In the original study[29], patients with ASD neglect the neutral intention of the actor in the accidental harm condition and rate the action as less morally acceptable compared to healthy controls. Although this might suggest an overreliance on action outcomes, it remains unclear why such alterations are not evident in other conditions as well (e.g., when autistic patients weigh the action-outcome stronger than the intention, they should rate attempted but not achieved harms as more morally acceptable, which they do not. For a detailed discussion on that topic see [53]). Accordingly, it might be that such tasks are suitable to detect alterations in ASD [29,54] and psychopathy [55] but are not sensitive enough to reveal the degree of alterations or the sub-aspect of ToM that is altered in SSD. Although ASD and SSD patients often reveal comparable behavioral ToM alterations[56], they are not necessarily caused by the same underlying malfunctioning mechanisms [57,58]. It is up to future replication studies to examine the sensitivity of this measure and to provide transdiagnostic analyses that will allow direct comparisons between clinical samples.

Although there is inconsistent evidence for a relation between antipsychotic medication and ToM performance in SSD [59,60], there remains the argument that behavioral alterations in SSD might be concealed by alleviating the effects of medication. To rule out an effect of antipsychotic medication on patients' moral judgments, we correlated medication (standardized Chlorpromazine equivalent in mg/day; [61]) to task performance showing that moral judgments did not vary with the degree of antipsychotic medication.

At last, the current findings might support the notion that moral judgments are not (or minimally) impaired in SSD. This is well in line with several recent studies that reveal minimal support for a significant impairment in moral judgment in SSD [14,17], especially when judging moral dilemmas from the self-perspective. Another counter for the conception of schizophrenic patients as morally insensitive is provided by neuroeconomic studies revealing that patients react in similar manners to unfair behavior towards others [18,19]. Nonetheless, the findings that intention-based moral judgments are presumably preserved in SSD is puzzling, given the countless evidence for impaired ToM processes on the behavioral [6,7,62] and neural [7] level and the altered empathy processes evident in our sample. Although empathy and ToM are both crucial for interactions and share some overlapping cortical areas, both processes also relate to distinct regions [63]. Furthermore, a recent meta-analysis showed that moral judgments are closer linked to ToM-related cortical areas than empathy-related areas which seemingly disapproves of a significant role of affect sharing during moral decision making [64]. Accordingly, our patient sample might suffer from decreased empathy while showing healthy-like moral judgments. Furthermore, patients with SSD can reveal altered cortical activation during moral decision making in ToM-related networks but simultaneously show no signs for altered behavioral moral judgments [17], which might account for the robust deficits identified in ToM studies [7,62] but preserved moral judgment in the current sample.

The current study supports previous evidence that SSD is not characterized by pronounced impairments in moral judgments, even when other people's intentions must be considered. This is well in line with recent meta-analyses showing that SSD patients do not engage in more immoral and violent behavior than the average person [65,66].

## 4.1 Limitations and outlook

Whereas neuroimaging and behavioral examinations of moral reasoning in psychiatric samples such as schizophrenia are undoubtedly important, they bear several pitfalls. First (and as

shortly mentioned in the introduction), findings on moral reasoning in schizophrenic patients are inconsistent and not all findings support the notion of impaired moral decision making or ToM in patients [18,19,67]. Second, although current research allows a better understanding of impaired behavioral outcomes and neuroimaging studies provide interesting insight into the cortical regions involved, most studies focus on certain sub-aspects of moral reasoning which are rarely put into a more exhaustive framework or model [23] and future studies must show how these sub-aspects integrate into a full model of morality in psychiatry.

Furthermore, we are yet far from an exhaustive understanding of how altered activation, connectivity or brain structure in psychiatric patients is related to behavioral outcome [68,69]. Whilst this is an ongoing issue not only in psychiatric research, the examination of morality bears a high responsibility, since it strongly influences the perception of how capable psychiatric patients are to understand their own actions. This in turn has an immediate influence on whether a psychiatric patient can be made responsible for his or her own actions which is an inevitable question when it comes to legal jurisdictions [68]. Taken together, there is a strong need for a more integrative account of morality in SSD, not only for research but also for a more sophisticated treatment of psychiatric patients in legal jurisdiction.

## Supporting information

**S1 Fig. Sequential analysis Accidental Harm.**
(TIF)

**S2 Fig. Sequential analysis Attempted Harm.**
(TIF)

**S3 Fig. Sequential analysis Intentional Harm.**
(TIF)

**S4 Fig. Sequential analysis Neutral Acts.**
(TIF)

**S1 File.**
(DOCX)

## Acknowledgments

We would like to thank Eva Reiter and Chelsea Hunt for their help with data acquisition and proofreading, respectively.

## Author Contributions

**Conceptualization:** Wolfgang Aichhorn, Martin Kronbichler.

**Formal analysis:** Lisa Kronbichler.

**Funding acquisition:** Martin Kronbichler.

**Investigation:** Lisa Kronbichler, Renate Stelzig-Schöler, Melanie Lenger, Stefanie Weber, Brandy-Gale Pearce, Luise-Antonia Reich.

**Methodology:** Lisa Kronbichler, Martin Kronbichler.

**Project administration:** Martin Kronbichler.

**Resources:** Wolfgang Aichhorn.

**Supervision:** Renate Stelzig-Schöler, Wolfgang Aichhorn, Martin Kronbichler.

**Visualization:** Lisa Kronbichler.

**Writing – original draft:** Lisa Kronbichler, Martin Kronbichler.

**Writing – review & editing:** Lisa Kronbichler, Martin Kronbichler.

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
