## [Decision Letter · Decision Letter 0]

13 Nov 2020

PONE-D-20-25364

Preserved Intention Understanding During Moral Judgments in Schizophrenia

PLOS ONE

Dear Dr. Kronbichler,

Thank you for submitting your manuscript to PLOS ONE. After careful consideration, we feel that it has merit but does not fully meet PLOS ONE’s publication criteria as it currently stands. Therefore, we invite you to submit a revised version of the manuscript that addresses the points raised during the review process.

Please could you pay careful attention to both reviewers' comments (please add ToM data on the subjects, if you have this, as recommended by Reviewer 1, if not it can be omitted). I also have some editorial comments as listed below: 

P3. I don’t think it is correct to say that social cognition is subsumed under theory of mind – most people would feel that the former is the broader construct.

P3. ‘Although there is evidence that patients show altered cortical activation during moral judgments [14], behavioral studies reveal mixed results showing fragile effects that often depend on question probe and perspective [13–16].

I don’ think it makes sense to juxtapose functional imaging and behavioural studies, especially as only one functional imaging study is cited. It might be better to have a paragraph in the discussion on functional imaging findings in healthy subjects and patients with schizophrenia.

P4. ‘Given that SSD patients have difficulties with belief understanding, we suggest that their ratings rely on the action outcome while neglecting the protagonist’s belief about his action.’

Please rephrase this sentence. If this is a hypothesis of the study, please indicate that this is the case. Alternatively, it might be better to remove it altogether.

P4-5. The paragraph on hypermentalizing would be better omitted, in my view. There has been almost no theoretical or experimental exploration of hypermentalizing in schizophrenia, and it is not clear what consequences it would have on a moral judgment task.

P5. ‘Participants in the patient group were 23 male adults, who had received a formal ICD-10 diagnosis (which was checked before study participation by certified psychiatrists)’

This statement does not meet accepted standards for meeting diagnostic criteria. Please expand on the process used. Please also move the section on exclusion criteria here.

Please separate the demographics table and the results table (which should be in a results section). Please move the text on empathy and cognition to before the table, so readers will know what the initials refer to. It is customary to give a measure like years of education or highest educational level achieved in a demographics table, so please include this if possible.

Please give considerably more detail on the Moral Judgements test, with at least one example. At the moment readers have to obtain and read reference 18 to know what the features of the test are. 

Please plan submit your revised manuscript in around three months or sooner. If you will need more time than this to complete your revisions, please reply to this message or contact the journal office at plosone@plos.org. Please include the following items when submitting your revised manuscript:

We look forward to receiving your revised manuscript.

Kind regards,

Peter John McKenna

Academic Editor

PLOS ONE

Journal Requirements:

2.) Please describe in your methods section how capacity to consent was determined for the participants in this study.

3.) Please include additional information regarding the vignettes used in the study and ensure that you have provided sufficient details that others could replicate the analyses. For instance, if you developed vignettes as part of this study and they is not under a copyright more restrictive than CC-BY, please include a copy, in both the original language and English, as Supporting Information. If materials, methods, and protocols are well established, authors may cite articles where those protocols are described in detail, but the submission should include sufficient information to be understood independent of these references (https://journals.plos.org/plosone/s/submission-guidelines#loc-materials-and-methods).

4.) We note that you have indicated that data from this study are available upon request. PLOS only allows data to be available upon request if there are legal or ethical restrictions on sharing data publicly. For more information on unacceptable data access restrictions, please see http://journals.plos.org/plosone/s/data-availability#loc-unacceptable-data-access-restrictions.

5.)Thank you for stating the following in the Acknowledgments Section of your manuscript:

[This work was supported by grants provided to Martin Kronbichler by the Austrian

315 Science Fund (FWF grant number: P 30390-B27) and the Scientific Funds of the Paracelsus

316 Medical University (grant number: E-13/18/097-KRO)]

 [The authors received no specific funding for this work.]

Reviewers' comments:

Reviewer's Responses to Questions

**Comments to the Author**

1. Is the manuscript technically sound, and do the data support the conclusions?

Reviewer #1: Partly

Reviewer #2: Yes

2. Has the statistical analysis been performed appropriately and rigorously? 

Reviewer #1: Yes

Reviewer #2: Yes

3. Have the authors made all data underlying the findings in their manuscript fully available?

Reviewer #1: Yes

Reviewer #2: No

4. Is the manuscript presented in an intelligible fashion and written in standard English?

Reviewer #1: Yes

Reviewer #2: Yes

5. Review Comments to the Author

Reviewer #1: Thank you for inviting me to review this interesting manuscript, investigating a relatively neglected area of social cognition research in schizophrenia at the intersection between psy- and phil-disciplines. The study investigates differences between patients with schizophrenia-spectrum disorder (SSD) and healthy controls on a moral judgment task which requires the understanding of others’ intentions. This is an original and novel contribution to the field, which however presents some shortcomings and, in my opinion, could be improved in relation to a number of issues summarised below:

1. KEYWORDS: not fully representative. I would suggest considering, for example: “schizophrenia, moral cognition, social cognition, theory of mind, empathy”.

2. ABSTRACT: Originality, relevance and need for this research are identified and the new task is briefly introduced, along with a clear summary of the conclusions. However, the sentence “we examine whether patients with SSD reveal alterations in moral judgments when the intentions of the acting person must be considered.” is slightly misleading and needs re-reading a few times in order to grasp what component of moral reasoning or what faculty the authors are trying to investigate (and therefore the research question is not clearly defined). It seems to me that they are conflating theory of mind and moral reasoning, and that the question they are actually asking is whether lacking theory of mind impacts on moral reasoning? Research question and aims could therefore be made clearer in the abstract.

3. INTRODUCTION: The introduction lacks a rigorous and critical conceptual engagement with the neurophilosophical and neuropsychological literature on the topic. The authors seem to take for granted the evidence for deficits in ability to appreciate other people’s beliefs and intentions in relation to schizophrenia-spectrum disorders across different contexts, whereas the literature is mixed and the effect of considering different components of moral reasoning, as well as different contexts, should be discussed. See for example (McCABE et al., 2004) (Carlson & Crockett, 2018; Yu et al., 2019) (Bluhm et al., 2015). Morality judgments might occur at the intersection between different dimensions of social cognition including for example moral decision‐making, judgment, and inference (Yu et al., 2019), while the authors’ reduce the discussion to the “neuronal level” (lines 53-55). In addition, the ToM deficits evidenced in ASD and schizophrenia are not necessarily similar on a phenomenological level (Stanghellini & Ballerini, 2011) and this should be discussed. Lines 62-65 refer to different paradigms (studies 12,17,13) but then the authors fail to explicate pros/cons, just re-directing the reader to (study 13) for an example. I suggest that the complexity and challenges of investigating ethical reasoning in psychiatric disorders should be more clearly delineated and that a more balanced overview of the neuropsychological literature should be provided.

4. METHODOLOGY: The authors provide a measure of overall empathy but do not provide any information on ToM skills in their sample. This is surprising given that in the introduction they focus on impairment both in ToM and empathy. Could their sample just have normal ToM? Other measures and statistics appear adequate but I cannot see any controlling for potential confounding factors.

5. RESULTS/DISCUSSION: Results are presented clearly and logically, and discussed in a stepwise fashion.

References:

Bluhm, R., Raczek, G., Broome, M., & Wall, M. B. (2015, July 16). Ethical Issues in Brain Imaging in Psychiatry. The Oxford Handbook of Psychiatric Ethics. https://doi.org/10.1093/oxfordhb/9780198732372.013.21

Carlson, R. W., & Crockett, M. J. (2018). The lateral prefrontal cortex and moral goal pursuit. Current Opinion in Psychology, 24, 77–82. https://doi.org/10.1016/j.copsyc.2018.09.007

McCABE, R., Leudar, I., & Antaki, C. (2004). Do people with schizophrenia display theory of mind deficits in clinical interactions? Psychological Medicine, 34(3), 401–412. https://doi.org/10.1017/S0033291703001338

Stanghellini, G., & Ballerini, M. (2011). What is it like to be a person with schizophrenia in the social world? A first-person perspective study on schizophrenic dissociality–Part 1: State of the art. Psychopathology, 44(3), 172–182.

Yu, H., Siegel, J. Z., & Crockett, M. J. (2019). Modeling Morality in 3-D: Decision-Making, Judgment, and Inference. Topics in Cognitive Science, 11(2), 409–432. https://doi.org/10.1111/tops.12382

Reviewer #2: Dear editors of PlosOne,

It has been a pleasure to review the manuscript by Kronbichler et al. entitled "Preserved Intention Understanding During Moral Judgments in Schizophrenia". This work deals with a clearly stated and interesting set of hypotheses and, in general, it has been correctly conducted. My comments are related to minor methodological aspects of the work. Specifically:

1.- The nature of the main analysis (three way, repeated measures ANOVA) should be stated in the Methods section (not in the results section as it is now).

2.- Exhaustive results from the non-bayesian ANOVA should be provided in a table.

3.- Results from post-hoc tests (one per each of the 2x2 conditions, comparing patients and controls) shouldn't be included in the demographics table (table 1) but in the ANOVA table (see point 1) or in a table of its own.

4.- Boxplots from Figure 1, clearly reveal that for some of the conditions (specially intended harm and neutral acts) there are some prominent floor and ceiling effects (observations gathered at extreme values) and net differences in variances between conditions, which, may violate the assumptions for parametric analyses.

Still, I understand that fitting a non-parametric version of a three way, repeated measures ANOVA may be difficult and unreliable, and we may consider results provided as valid. However, at least within condition two group comparisons (post-hoc t-test) should be supplemented with results from non-parametric tests (ideally permutation tests) .

5.- The inclusion of Bayesian results is also informative and reinforces the classic ANOVA outcome (both analyses agree in broad terms) however, due to the non-familiarity of some of the readers with Bayesian methods, a clear explanation on the different statistics reported (mainly Bayes Factors) should be included in the text.

6.- Finally, Table captions are very poor and should be improved by including extensive information on tables contents.

6. PLOS authors have the option to publish the peer review history of their article (what does this mean?). If published, this will include your full peer review and any attached files.

Reviewer #1: No

Reviewer #2: No

---

## [Author Response · Author response to Decision Letter 0]

8 Apr 2021

Dear Reviewers, dear Editor,

Thank you so much for putting your valuable time into this review. We hope that our changes suit your expectations and we sincerely want to thank you for helping to improve our manuscript. The list of changes is attached below in the initial order of the review.

Editor

P3. I don’t think it is correct to say that social cognition is subsumed under theory of mind – most people would feel that the former is the broader construct.

Thank you, of course you are right, the wording of this sentence was misleading. We rewrote this sentence, describing TOM as a (sub)factor or social cognition, which better fits the common description of TOM.

P3. ‘Although there is evidence that patients show altered cortical activation during moral judgments [14], behavioral studies reveal mixed results showing fragile effects that often depend on question probe and perspective [13–16].

● I don’ think it makes sense to juxtapose functional imaging and behavioural studies, especially as only one functional imaging study is cited. It might be better to have a paragraph in the discussion on functional imaging findings in healthy subjects and patients with schizophrenia.

We rewrote the introduction and avoided a direct comparison between neuroimaging and behavioral data since juxtaposing behavioral and neuroimaging data is misleading. Since we nonetheless provide information on both data types, we now explicitly state in the discussion that neuroimaging does not directly map on behavior.

P4. ‘Given that SSD patients have difficulties with belief understanding, we suggest that their ratings rely on the action outcome while neglecting the protagonist’s belief about his action.’

● Please rephrase this sentence. If this is a hypothesis of the study, please indicate that this is the case. Alternatively, it might be better to remove it altogether.

Since this is one of our assumptions about the task, we rephrased the sentence. It now reads: ‘If our SSD sample has difficulties understanding the intention and belief of the story protagonist, we should find that patients mostly rely their ratings on the action outcome since the intention of the protagonist (to harm or not to harm the other person) is not informative to them.’

P4-5. The paragraph on hypermentalizing would be better omitted, in my view. There has been almost no theoretical or experimental exploration of hypermentalizing in schizophrenia, and it is not clear what consequences it would have on a moral judgment task.

This section was mostly added for the sake of completeness - I removed it.

P5. ‘Participants in the patient group were 23 male adults, who had received a formal ICD-10 diagnosis (which was checked before study participation by certified psychiatrists)’

● This statement does not meet accepted standards for meeting diagnostic criteria. Please expand on the process used. 

Patients were examined using the Mini-International Neuropsychiatric Interview (Sheehan et al. 1998) This is now stated in the Participant section.

● Please also move the section on exclusion criteria here.

Exclusion criteria were moved upwards.

Please separate the demographics table and the results table (which should be in a results section). Please move the text on empathy and cognition to before the table, so readers will know what the initials refer to. It is customary to give a measure like years of education or highest educational level achieved in a demographics table, so please include this if possible.

Demographics and results are now presented in separate Tables (T1,T2). Education level was added to Table 1.

Please give considerably more detail on the Moral Judgements test, with at least one example. At the moment readers have to obtain and read reference 18 to know what the features of the test are. 

The task description was rewritten and is now described using an example story. We hope that this facilitates the understanding of this task. Furthermore, two complete stories are now added to the supplementary material.

2.) Please describe in your methods section how capacity to consent was determined for the participants in this study.

Thank you for pointing this out, since this is an extremely important topic. Patient capacity to participate (and give informed consent) was determined in two steps: First, only patients authorized to give informed consent were taken into account. Patients under external custody (either due to their age or their mental state) were not included. Furthermore, all patients were recruited by their treating clinical psychiatrists, and only patients whose mental state was considered stable enough were recruited. There were no exceptions to this procedure. 

The methods section now contains this statement: Only participants who were authorized to give informed consent themselves (>18 years, not under external custody) were included in the study. Furthermore, patient recruitment was exclusively done by an experienced team of psychiatrists. 

3.) Please include additional information regarding the vignettes used in the study and ensure that you have provided sufficient details that others could replicate the analyses. For instance, if you developed vignettes as part of this study and they is not under a copyright more restrictive than CC-BY, please include a copy, in both the original language and English, as Supporting Information. If materials, methods, and protocols are well established, authors may cite articles where those protocols are described in detail, but the submission should include sufficient information to be understood independent of these references (https://journals.plos.org/plosone/s/submission-guidelines#loc-materials-and-methods).

The test material is well established (Moran et al., 2011) and described in detail there. To nonetheless get a better idea of how the stories are designed, we added the english and german versions of two example vignettes in the supplementary material.

4.) We note that you have indicated that data from this study are available upon request. PLOS only allows data to be available upon request if there are legal or ethical restrictions on sharing data publicly. For more information on unacceptable data access restrictions, please see http://journals.plos.org/plosone/s/data-availability#loc-unacceptable-data-access-restrictions.

The following statement is also included in the cover letter:

 Currently, we do not have approval from the local ethical committee responsible for this study (Ethikkommission des Landes Salzburg - Ethics committee of the regional state of Salzburg) to publicly share single patients data of any sort, without the explicit consent of patients to share their individual data publicly. Since our study consent form did not include such a clause, we are not allowed to make single-subject data publicly available. For data access, please write an email to the corresponding author, which will clarify the potential to share the data with the local ethics committee. We are currently trying to obtain a more general allowance from the ethics committee for sharing at least the single subject behavioural data but this process will take a while. If a more general data sharing allowance is obtained, we will update the information with the article and share the data on a public repository (if allowed).

Ethikkommission für das Bundesland Salzburg

Sebastian-Stief-Gasse 2

5020 Salzburg

Contact:

Mag. Ulrike Wendl-Toiflhart

Tel.: +43-(0)662-8042-2375

E-Mail: ethikkommission@salzburg.gv.at

5.)Thank you for stating the following in the Acknowledgments Section of your manuscript:

This work was supported by grants provided to Martin Kronbichler by the Austrian

315 Science Fund (FWF grant number: P 30390-B27) and the Scientific Funds of the Paracelsus

316 Medical University (grant number: E-13/18/097-KRO).

 [The authors received no specific funding for this work.]

Thank you for pointing this out. Please move below section to the Funding Statement:

This work was supported by grants provided to Martin Kronbichler by the Austrian Science Fund (FWF grant number: P 30390-B27) and the Scientific Funds of the Paracelsus Medical University (grant number: E-13/18/097-KRO).

Reviewer #1: 

Thank you for inviting me to review this interesting manuscript, investigating a relatively neglected area of social cognition research in schizophrenia at the intersection between psy- and phil-disciplines. The study investigates differences between patients with schizophrenia-spectrum disorder (SSD) and healthy controls on a moral judgment task which requires the understanding of others’ intentions. This is an original and novel contribution to the field, which however presents some shortcomings and, in my opinion, could be improved in relation to a number of issues summarised below:

1. KEYWORDS: not fully representative. I would suggest considering, for example: “schizophrenia, moral cognition, social cognition, theory of mind, empathy”.

Keywords were changed accordingly. The term “schizophrenia” was omitted since it is already present in the title.

2. ABSTRACT: Originality, relevance and need for this research are identified and the new task is briefly introduced, along with a clear summary of the conclusions. However, the sentence “we examine whether patients with SSD reveal alterations in moral judgments when the intentions of the acting person must be considered.” is slightly misleading and needs re-reading a few times in order to grasp what component of moral reasoning or what faculty the authors are trying to investigate (and therefore the research question is not clearly defined). It seems to me that they are conflating theory of mind and moral reasoning, and that the question they are actually asking is whether lacking theory of mind impacts on moral reasoning? Research question and aims could therefore be made clearer in the abstract.

This section of the abstract was rewritten in order to make our research interest clearer.

3. INTRODUCTION: The introduction lacks a rigorous and critical conceptual engagement with the neurophilosophical and neuropsychological literature on the topic. The authors seem to take for granted the evidence for deficits in ability to appreciate other people’s beliefs and intentions in relation to schizophrenia-spectrum disorders across different contexts, whereas the literature is mixed and the effect of considering different components of moral reasoning, as well as different contexts, should be discussed. See for example (McCABE et al., 2004) (Carlson & Crockett, 2018; Yu et al., 2019) (Bluhm et al., 2015). 

Morality judgments might occur at the intersection between different dimensions of social cognition including for example moral decision‐making, judgment, and inference (Yu et al., 2019), while the authors’ reduce the discussion to the “neuronal level” (lines 53-55). In addition, the ToM deficits evidenced in ASD and schizophrenia are not necessarily similar on a phenomenological level (Stanghellini & Ballerini, 2011) and this should be discussed. Lines 62-65 refer to different paradigms (studies 12,17,13) but then the authors fail to explicate pros/cons, just re-directing the reader to (study 13) for an example. I suggest that the complexity and challenges of investigating ethical reasoning in psychiatric disorders should be more clearly delineated and that a more balanced overview of the neuropsychological literature should be provided.

We mostly rewrote the introduction to provide a more sophisticated overview of the currently existing literature. We found that some of your suggestions would better fit in the discussion section of the paper, therefore we moved aspects like the link between neuroimaging and behavioral data there. We hope this is alright, since we wanted to incorporate all the valuable suggestions made by the reviewer.

4. METHODOLOGY: The authors provide a measure of overall empathy but do not provide any information on ToM skills in their sample. This is surprising given that in the introduction they focus on impairment both in ToM and empathy. Could their sample just have normal ToM? Other measures and statistics appear adequate but I cannot see any controlling for potential confounding factors.

Unfortunately, data on TOM skills are not available for this sample.

Correlation analyses are (and were in the first manuscript) reported examining the influence of symptom severity, medication and illness duration. We politely refer to paragraph 2 in the results section: Pearson Correlations indicate that symptom severity (PANSS) and medication (Chlorpromazine equivalent) were not related to task performance (rs < .38, ps > .09). Illness duration was moderately related to permissibility ratings in Intended Harm scenarios (r = .41, p = .052). 

5. RESULTS/DISCUSSION: Results are presented clearly and logically, and discussed in a stepwise fashion.

Reviewer #2: 

Dear editors of PlosOne,

It has been a pleasure to review the manuscript by Kronbichler et al. entitled "Preserved Intention Understanding During Moral Judgments in Schizophrenia". This work deals with a clearly stated and interesting set of hypotheses and, in general, it has been correctly conducted. My comments are related to minor methodological aspects of the work. Specifically:

1.- The nature of the main analysis (three way, repeated measures ANOVA) should be stated in the Methods section (not in the results section as it is now).

The description is moved to the methods section.

2.- Exhaustive results from the non-bayesian ANOVA should be provided in a table.

Descriptives, results of the ANOVA and t-test are now listed in Table 2.

3.- Results from post-hoc tests (one per each of the 2x2 conditions, comparing patients and controls) shouldn't be included in the demographics table (table 1) but in the ANOVA table (see point 1) or in a table of its own.

See point above.

4.- Boxplots from Figure 1, clearly reveal that for some of the conditions (specially intended harm and neutral acts) there are some prominent floor and ceiling effects (observations gathered at extreme values) and net differences in variances between conditions, which, may violate the assumptions for parametric analyses.

Still, I understand that fitting a non-parametric version of a three way, repeated measures ANOVA may be difficult and unreliable, and we may consider results provided as valid. However, at least within condition two group comparisons (post-hoc t-test) should be supplemented with results from non-parametric tests (ideally permutation tests) .

Thank you sincerely for pointing this out. For the sake of simplicity (we felt the results section is already pretty long), we added the permutation test results in the supplementary material and mentioned their similar outcome in our results section.

5.- The inclusion of Bayesian results is also informative and reinforces the classic ANOVA outcome (both analyses agree in broad terms) however, due to the non-familiarity of some of the readers with Bayesian methods, a clear explanation on the different statistics reported (mainly Bayes Factors) should be included in the text.

Thank you for pointing this out. Although quite informative, the idea behind bayes statistics is not (yet) as popular as conventional NHST. Therefore, we added a short paragraph (2.5) where we describe the meaning of bayes factors and refer to practical guides for a more in depth description.

6.- Finally, Table captions are very poor and should be improved by including extensive information on tables contents.

Table captions were rewritten and provide more information now.

---

## [Editor Report · Decision Letter 1]

14 Apr 2021

PONE-D-20-25364R1

Preserved intention understanding during moral judgments in schizophrenia

PLOS ONE

Dear Dr. Kronbicher,

Thank you for submitting your revised manuscript to PLOS ONE. It is now nearly ready for acceptance but there are few minor issues that need addressing:

1. The article is understandable but the use of English has frequent grammatical errors, particularly with regard to  tense. If you were able to have a native English speaker review the text this would be helpful.

2. P7. 'The study was part of a longitudinal study including behavioral and MRI acquisitions in affective and psychotic disorders approved by the local ethics committee'. This sentence is unnecessary and confusing, as you have already stated that the study had ethics committee approval - please remove or amend.

3. P17. You begin a paragraph of the discussion with 'Probably the mainstream opener when discussing a null effect...'. It would be better to replace this with something like 'Probably the most important consideration for a null effect...'.

4. Please incorporate the two footnotes into the main body of the text.

We look forward to receiving your revised manuscript.

Kind regards,

Peter John McKenna

Academic Editor

PLOS ONE
---

## [Author Response · Author response to Decision Letter 1]

20 Apr 2021

Dear Editorial Management,

in the following, I'd like to address the issues raised in your last email:

1) the manuscript was given to a british colleague for proofreading

2) the sentence has been removed

3) the sentence was changed according to your suggestion

4) the footnotes were incorporated in the main text

Note: Could you please change the name and affiliation of melanie LENGER, since there was an error in the initial version of the manuscript. I changed it in the draft but I was not able to change it in the online system.

thank you very much,

sincerely, lisa kronbichler

---

## [Editor Report · Decision Letter 2]

22 Apr 2021

Preserved intention understanding during moral judgments in schizophrenia

PONE-D-20-25364R2

Dear Dr. Kronbichler,

We’re pleased to inform you that your manuscript has been judged scientifically suitable for publication and will be formally accepted for publication once it meets all outstanding technical requirements.

Kind regards,

Peter John McKenna

Academic Editor

PLOS ONE
---

## [Editor Report · Acceptance letter]

26 Apr 2021

PONE-D-20-25364R2 

Preserved intention understanding during moral judgments in schizophrenia 

Dear Dr. Kronbichler:

I'm pleased to inform you that your manuscript has been deemed suitable for publication in PLOS ONE. Congratulations! Your manuscript is now with our production department. 

Kind regards, 

on behalf of

Dr. Peter John McKenna 

Academic Editor

PLOS ONE